

# Transcriptome response to elevated atmospheric $CO_2$ concentration in the Formosan subterranean termite, *Coptotermes formosanus* Shiraki (Isoptera: Rhinotermitidae)

Wenjing Wu[1], Zhiqiang Li[1], Shijun Zhang[1], Yunling Ke[1] and Yahui Hou[1,2]

[1] Guangdong Key Laboratory of Integrated Pest Management in Agriculture, Guangdong Public Laboratory of Wild Animal Conservation and Utilization, Guangdong Institute of Applied Biological Resources, Guangzhou, Guangdong, China

[2] College of Forestry, Northeast Forestry University, Harbin, Heilongjiang, China

## ABSTRACT

**Background**. Carbon dioxide ($CO_2$) is a pervasive chemical stimulus that plays a critical role in insect life, eliciting behavioral and physiological responses across different species. High $CO_2$ concentration is a major feature of termite nests, which may be used as a cue for locating their nests. Termites also survive under an elevated $CO_2$ concentration. However, the mechanism by which elevated $CO_2$ concentration influences gene expression in termites is poorly understood.

**Methods**. To gain a better understanding of the molecular basis involved in the adaptation to $CO_2$ concentration, a transcriptome of *Coptotermes formosanus* Shiraki was constructed to assemble the reference genes, followed by comparative transcriptomic analyses across different $CO_2$ concentration (0.04%, 0.4%, 4% and 40%) treatments.

**Results**. (1) Based on a high throughput sequencing platform, we obtained approximately 20 GB of clean data and revealed 189,421 unigenes, with a mean length and an N50 length of 629 bp and 974 bp, respectively. (2) The transcriptomic response of *C. formosanus* to elevated $CO_2$ levels presented discontinuous changes. Comparative analysis of the transcriptomes revealed 2,936 genes regulated among 0.04%, 0.4%, 4% and 40% $CO_2$ concentration treatments, 909 genes derived from termites and 2,027 from gut symbionts. Genes derived from termites appears selectively activated under 4% $CO_2$ level. In 40% $CO_2$ level, most of the down-regulated genes were derived from symbionts. (3) Through similarity searches to data from other species, a number of protein sequences putatively involved in chemosensory reception were identified and characterized in *C. formosanus*, including odorant receptors, gustatory receptors, ionotropic receptors, odorant binding proteins, and chemosensory proteins.

**Discussion**. We found that most genes associated with carbohydrate metabolism, energy metabolism, and genetic information processing were regulated under different $CO_2$ concentrations. Results suggested that termites adapt to ~4% $CO_2$ level and their gut symbionts may be killed under high $CO_2$ level. We anticipate that our findings provide insights into the transcriptome dynamics of $CO_2$ responses in termites and form the basis to gain a better understanding of regulatory networks.

Corresponding author
Zhiqiang Li, lizq@giabr.gd.cn

## INTRODUCTION

Despite the low concentration of carbon dioxide ($CO_2$) in air, it plays a critical role in insect life. Insects not only live in the normal atmosphere environment, but are also sometimes exposed to higher or lower $CO_2$ concentrations. Naturally high $CO_2$ concentration is likely to occur in holes under the bark of trees or stumps, in the soil when it is covered by ice and snow, or inside decomposing organic matter. Fluctuations of atmospheric $CO_2$ could evoke behavioral and physiological responses in insects. On the one hand, $CO_2$ acts as an attractive cue to elicit behavioral responses in many insects, such as seeking food and hosts, avoiding conspecifics, and locating nests (*Guerenstein & Hildebrand, 2008*). For example, mosquitoes depend on $CO_2$ to locate human hosts whose volatile emissions contain $CO_2$ (*Gillies, 1980*; *Guerenstein & Hildebrand, 2008*). Many moths measure the $CO_2$ gradients, which indicate the floral quality, to find more and better nectar (*Guerenstein et al., 2004*; *Thom et al., 2004*). In *Drosophila*, high concentrations of $CO_2$ elicit an avoidance response to other individuals (*Suh et al., 2004*). Social insects such as bees, wasps, ants and termites may detect $CO_2$ concentration to locate their nests, in which $CO_2$ concentration is much higher than the atmospheric concentration (*Seeley, 1974*). On the other hand, physiological effects of $CO_2$ are diverse. In the nervous system, increasing $CO_2$ concentration induce sub-lethal or lethal effects (*Nicolas & Sillans, 1989*). In the respiratory and circulatory system, changes in $CO_2$ regulate the opening of the spiracles. In developmental processes, high $CO_2$ may decrease metabolic rates, reduce weight, affect size, or prolong larval life and growth. In regards to reproduction, $CO_2$ may delay or impede mating activity, accelerate oviposition, or stimulate vitellogenin synthesis (*Nicolas & Sillans, 1989*).

Termites contribute up to 2% of the natural efflux of $CO_2$ from terrestrial sources (*Sugimoto, Bignell & MacDonald, 2000*) and 10% from the soil surface (*De Gerenyu et al., 2015*). High $CO_2$ concentration is a major feature of termite nests. Inside the nests, termite activity takes place under an elevated $CO_2$ concentration (0.3–5%) and occasionally up to 15%, but outside the nests, termites are exposed to the natural $CO_2$ concentration in air (approximately 0.04%) (*Ziesmann, 1996*). It is suggested that $CO_2$ concentration may provide information on location of termite nests. *Bernklau et al. (2005)* reported that *Reticulitermes* spp. were attracted to $CO_2$ concentrations between 5 and 50 mmol/mol and $CO_2$ could be used as an attractant in baiting systems to elicit termites to an insecticide. This finding has been commercialized and is used in Ensystex bait systems under the name Focus.

The chemosensory system is usually used by insects to sense odorants, the taste of food, or other chemical stimuli in the environment. Sensory structures for detecting changes in atmospheric $CO_2$ have been identified and described in Lepidoptera, Diptera, Hymenoptera, and Isoptera (*Stange & Stowe, 1999*). The structures typically contain clusters of wall-pore type sensilla and are housed in pits or capsules. In different insects,

they are located on either the palps (moths, mosquitoes, flies, and beetles) or the antennae (bees, ants, and termites) (*Stange & Stowe, 1999*). In termites, study of *Schedorhinotermes lamanianus* showed that sensory cells in the antennal sensilla may be sensitive to both $CO_2$ and odorant (*Ziesmann, 1996*). The insect chemosensory proteins are various and mainly located in the sensory structures, such as odorant receptor (OR), gustatory receptor (GR), ionotropic receptor (IR), odorant binding protein (OBP), and chemosensory protein (CSP) families. Several studies have aimed to elucidate their underlying mechanisms and functions. The first study on the molecular bases of $CO_2$ reception was in *Drosophila*. Two *GR* genes (*GR21a* and *GR63a*) were identified, and co-expression of them was necessary to confer a $CO_2$ response (*Jones et al., 2007*; *Kwon et al., 2007*). Orthologues of *GR21a* and *GR63a* have been identified in butterfly, moth, beetle, mosquito, and termite species, but not in honeybees, pea aphids, ants, locusts and wasps (*Xu & Anderson, 2015*). These genomic differences may suggest different chemoreceptors and mechanisms for $CO_2$ detection among different insects.

The objective of this study was to investigate the effects of elevated $CO_2$ concentrations on the Formosan subterranean termite (*Coptotermes formosanus* Shiraki) in artificial, sealed chambers in the laboratory. Lower termite *C. formosanus* is among the most destructive species worldwide and characterized by the dependence on protozoan symbionts for cellulose digestion. In the present study, to enable comprehensive gene expression profiling, we generated as complete a reference transcriptome as possible for *C. formosanus*. Pooled RNA from different developmental stages and castes was used as starting material for Illumina sequencing. Next, we constructed four libraries of *C. formosanus* workers at different $CO_2$ concentrations and compared gene expression profiles among them. We identified differentially expressed genes, analyzed sensitive processes that were involved in the response to elevated $CO_2$, and screened genes associated with the chemosensory system. These assembled and annotated transcriptome sequences will facilitate gene discovery in *C. formosanus* and functional analysis of expressed genes and deepen our understanding of the molecular basis of responses to elevated $CO_2$ concentrations in termites and other insects.

## MATERIALS & METHODS

### Insects and $CO_2$ treatments

Colonies of *C. formosanus* termites, collected in Guangzhou International Biotech Island (23°04′01.71″N, 113°21′47.74″E), Guangdong, China, were kept in the laboratory in 5.0-L covered plastic boxes containing blocks of pine wood in 85 ± 5% humidity at 27 ±1 °C until they were used in experiments. No specific permissions were required for accessing these locations for sampling activities, and no endangered or protected species were involved in the study.

To comprehensively investigate the differences in gene expression when $CO_2$ concentration was elevated, we performed comparative transcriptome analysis among worker termites rearing at 0.04% $CO_2$ (natural $CO_2$ level), 0.4% $CO_2$ (low $CO_2$ level), 4% $CO_2$ (medium $CO_2$ level), and 40% $CO_2$ (high $CO_2$ level). $CO_2$ treatments were performed

in gastight containers, which rinsed with distilled water. One hundred termite workers and ten soldiers were placed in each container with moistened sterile vermiculite (Hoffman, Landsville, PA) and a filter paper disc (8 cm in diameter). Different concentrations of $CO_2$, 0.04%, 0.4%, 4% and 40% were achieved by inputting gas mixtures of 0.04%, 0.4%, 4% and 40% $CO_2$; 21% $O_2$; and the balance $N_2$. $CO_2$ concentrations were confirmed using a $CO_2$ sensor (Type-IR- $CO_2$ gas tester, Heraeus), with accuracy range of 0–1% $\pm0.05\%$ $CO_2$ absolute; 1–25% $\pm5\%$ $CO_2$ of reading; 25–60% $\pm10\%$ $CO_2$ of reading. At a substantially constant temperature (27 $\pm1$ °C) and humidity (85 $\pm5\%$), all treatment groups were exposed for 72 hr and then collected live worker termites.

## Sampling and RNA extraction

For collecting samples of RNA, untreated individuals (including the worker, soldier and reproductive castes) of *C. formosanus* from our laboratory were collected and frozen immediately in liquid nitrogen and stored in −80 °C freezers until use. The samples of termites were randomly chosen with development stages, including larva, worker, pre-soldier, and soldier. Each sample containing 50 whole body individuals from each caste and stage was subjected to RNA isolation. Samples of 50 live workers from each $CO_2$ treatment were also collected and frozen immediately in liquid nitrogen and stored in −80 °C. Total RNA was extracted using the RNAsimple Total RNA Kit (TIANGEN, Beijing, China) according to the manufacturer's instructions. RNA quantity and quality were assessed using the NanoDrop spectrophotometer (Nanodrop Technologies Inc., Rockland, DE, USA) and the Agilent 2100 Bioanalyzer (Santa Clara, CA, USA). The standards applied were $OD_{260}/OD_{230} \geq 1.8$, $1.8 \leq OD_{260}/OD_{280} \leq 2.2$, and RNA integrity number values >8.0. RNA samples were used for cDNA library construction and qRT-PCR.

## cDNA library construction and sequencing

For reference transcriptome of *C. formosanus*, equal amounts of RNA from untreated samples (larva, worker, pre-soldier, soldier, and reproductive) and all $CO_2$-treated samples were mixed, designated as Cfo. For transcriptomic comparison among $CO_2$ treatments, RNA from 0.04%, 0.4%, 4%, and 40% $CO_2$-treated workers were used, designated as T1, T2, T3, and T4, respectively. T1 was served as the control group. Finally, five library constructions (Cfo, T1, T2, T3, and T4) and the RNA sequencing were performed by the Biomarker Biotechnology Corporation (Beijing, China). Approximately, 5 μg of total RNA for each sample was used for the construction of libraries using TruSeq Stranded mRNA Sample Prep Kit (Illumina Inc., San Diego, CA, USA) according to the manufacturer's protocol. Sequencing was performed in a v3 flowcell on an Illumina HiSeq$^{TM}$ 2500 sequencer, using the TruSeq PE Cluster Kit v3 (Illumina PE-401-3001) and the TruSeq SBSHS Kit v3 200 cycles (Illumina FC-401-3001).

## *De novo* transcriptome assembly and annotation

Raw reads were filtered by removing the adaptor sequences, empty reads and low quality sequences (reads with more than 50% of bases with $Q$-value $\leq 20$). The clean reads were then assembled *de novo* using the Trinity platform (http://trinityrnaseq.github.io) with the parameters of 'K-mer $= 25$, group pairs distance $= 300$' (*Grabherr et al., 2011*). By

performing pair-end joining and gap filling, contigs were assembled into transcripts, and the longest copy of redundant transcripts was regarded as a unigene (*Grabherr et al., 2011*; *Haas et al., 2013*).

The obtained unigenes were compared against public databases, including NCBI non-redundant nucleotide sequence (NT) database using BLASTn (version 2.2.14), NCBI non-redundant protein (NR), Swiss-Prot, Kyoto Encyclopedia of Genes and Genomes (KEGG), Clusters of Orthologous Group (COG), euKaryotic Orthologous Group (KOG), and Protein family (PFAM) databases using BLASTx (version 2.2.23) with an E-value cutoff at $10^{-5}$ (*Kanehisa et al., 2004*; *Koonin et al., 2004*; *Tatusov et al., 2000*). To identify Gene Ontology (GO) terms describing biological processes, molecular functions, and cellular components, the Swiss-Prot BLAST results were imported into Blast2GO 3.0.8 (*Götz et al., 2008*).

## Analysis of gene expression and identification of differentially expressed genes (DEGs)

The abundance of all genes was normalized and calculated by RSEM (*Li & Dewey, 2011*) and represented by the fragments per kilo base of transcript per million mapped reads (FPKM) value (*Trapnell et al., 2010*). We kept transcript isoform predictions whose FPKM > 0.03. DEGs were identified using EBSeq with conditions of FDR (False Discovery Rate) <0.01 and fold-change ≥2 (*Leng et al., 2013*). GO enrichment analysis of DEGs was implemented by using the Bioconductor package topGO (available at http://www.bioconductor.org/packages/release/bioc/html/topGO.html). Kolmogorov–Smirnov (KS) test was used to test the enrichment of GO terms with DEGs at a significance level of $P \leq 0.05$ (*Alexa & Rahnenführer, 2009*). For the pathway enrichment analysis, we mapped all DEGs to terms in the KEGG database and looked for significantly enriched KEGG terms compared to the transcriptome database. We used KEGG Automatic Annotation Server (http://www.genome.jp/tools/kaas/) with the parameters of search program = 'BLAST,' GENEs data set = 'for Eukaryotes, including auto-selected organisms and all insect organisms,' and assignment method = 'BBH (bi-directional best hit).'

## Validation of RNA-Seq data

To confirm the differential expression of genes revealed by RNA-Seq, 10 DEGs were chosen for quantitative real-time PCR (qRT-PCR) validation. qRT-PCR was performed by using the TaKaRa SYBR® Premix Ex Taq Perfect Real Time qPCR Kit (TaKaRa, Japan) and the StrataGene Mx3000P QPCR System (Agilent Technologies, Santa Clara, CA, USA). For each gene, 100 ng of total RNA was used as a template in a mixture of specific primers (10 μM) (Table S1) and Master Mix to a final volume of 20 μL following manufacturer's instruction. The qRT-PCR program was set to: 95 °C for 30 s of pre-incubation, 40 cycles of 95 °C for 5 s, 60 °C for 30 s, and 72 °C for 30 s of amplification. The specificity of the PCR products from each primer pair was confirmed by melting-curve analysis and agarose gel electrophoresis. Three biological replicates of each treatment were tested. All measurements were performed in triplicate. 18S ribosomal RNA was used as a reference gene to normalize gene expression according to previous study (*Zhang et al., 2011*). In

**Table 1   Summary of the Illumina sequencing and Trinity assembly.**

| Sample | Cfo | T1 | T2 | T3 | T4 |
|---|---|---|---|---|---|
| Raw bases (Gb) | 11.16 | 2.34 | 2.21 | 2.27 | 2.24 |
| Adapter (%) | 0.23 | 0.57 | 0.65 | 0.28 | 0.36 |
| rRNA (%) | 0.91 | 0.7 | 0.57 | 0.66 | 0.47 |
| Clean reads | 11,019,041,624 | 2,301,858,049 | 2,172,585,864 | 2,237,528,075 | 2,214,078,397 |
| Clean bases (Gb) | 11.02 | 2.3 | 2.17 | 2.24 | 2.21 |
| GC content (%) | 45.37 | 44.6 | 43.97 | 44.99 | 44.52 |
| Q30 (%) | 88.72 | 88.12 | 87.98 | 88.05 | 88.28 |
| Total number of contigs | 11,970,460 | 1,938,383 | 1,699,673 | 1,787,590 | 1,716,315 |
| Total number of transcripts | 316,037 | 97,565 | 87,253 | 89,963 | 87,169 |
| Total number of unigenes | 189,421 | 74,029 | 68,103 | 69,636 | 65,083 |
| Unigene length of 200–300 bp | 84,246 (44.48%) | 30,789 (41.59%) | 27,735 (40.73%) | 28,274 (40.60%) | 26,333 (40.46%) |
| Unigene length of 300–500 bp | 50,522 (26.67%) | 19,843 (26.80%) | 18,405 (27.03%) | 18,627 (26.75%) | 17,217 (26.45%) |
| Unigene length of 500–1,000 bp | 30,559 (16.13%) | 12,828 (17.33%) | 11,880 (17.44%) | 12,351 (17.74%) | 11,349 (17.44%) |
| Unigene length of 1,000–2,000 bp | 13,789 (7.28%) | 6,464 (8.73%) | 6,157 (9.04%) | 6,292 (9.04%) | 5,974 (9.18%) |
| Unigene length of >2,000 bp | 10,303 (5.44%) | 4,105 (5.55%) | 3,926 (5.76%) | 4,092 (5.88%) | 4,210 (6.47%) |
| Total length (bp) of unigenes | 119,236,672 | 46,419,882 | 43,644,839 | 44,699,752 | 43,186,051 |
| N50 length (bp) of unigenes | 974 | 956 | 994 | 996 | 1,080 |
| Mean length (bp) of unigenes | 629 | 627 | 641 | 642 | 664 |

addition, the expression of 18S rRNA in RNA-seq and preliminary qPCRs using the $CO_2$-treated workers was stable (Fig. S1). The $2^{-\Delta\Delta Ct}$ method was used to analyze the qRT-PCR data and assign relative expression differences (*Livak & Schmittgen, 2001*).

## Availability of supporting data

All sequence data have been submitted to GenBank Sequence Read Archive databases under accession number SRP068272 and SRP068332, and associated with Bioproject PRJNA308390 and PRJNA308507, respectively. Their accessions are SRR3095926 for Cfo (reference transcriptome of *C. formosanus*), SRR3097983 for T1, SRR3097984 for T2, SRR3097985 for T3, and SRR3097987 for T4.

## RESULTS

### Transcriptome sequencing and assembly

An overview of the sequencing and assembly is outlined in Table 1. After quality control, the number of clean bases in the reference transcriptome of *C. formosanus*, and four $CO_2$ treatments T1, T2, T3, and T4 were 11.02, 2.30, 2.17, 2.24 and 2.21 GB, respectively, with an average GC content of 44.69% and a Q30 of 88.23% (Table 1). After assembly, 316,037 transcripts were completed and assembled into 189,421 unigenes. Many unigenes had a

length between 200–1,000 bp. The mean length and N50 (50% of the transcriptome is in unigenes of this size or larger) length of unigenes were 629 bp and 974 bp, respectively. A larger N50 length and mean length are considered indicative of better assembly (*Garg et al., 2011*).

## Functional annotation and classification

After annotation, the number of unigenes with different length annotated in different databases and their percentage were counted (Table S2). The NR database (61,407, 32.42%) had the largest match. The Swiss-Prot (35,633, 18.81%), PFAM (32,444, 17.13%), and KOG (30,531, 16.12%) shared similar quantities. Unigene length over 1,000 bp annotated more successfully than length less than 1,000 bp (Table S2).

Totally 16,552 unigenes were annotated into 55 sub-categories belonging to three main GO categories: biological process (BP), cellular component (CC), and molecular function (MF) (Fig. S2). There were 20, 19, and 16 sub-categories in BP, CC, and MF, respectively. The top sub-categories were metabolic process (10,208 unigenes), cell part (4,100 unigenes), and catalytic activity (9,975 unigenes) in BP, CC, and MF, respectively. By KOG classifications, 30,531 unigenes were classified functionally into 25 categories. The cluster of 'signal transduction mechanisms' was the largest group, which had 6,631 unigenes. Pathway analyses were also performed on all assembled unigenes to understand the biological functions of genes and how these genes interact. A total of 16,444 unigenes were functionally classified into five KEGG categories (Fig. S3): genetic information processing (5,403 unigenes, 788 enzymes), metabolism (2,169 unigenes, 487 enzymes), cellular processes (2,146 unigenes, 358 enzymes), environmental information processing (1,235 unigenes, 218 enzymes), and organismal systems (548 unigenes, 90 enzymes). Among 19 sub-categories, 'translation,' 'transport and catabolism,' and 'folding, sorting and degradation' were the top three sub-categories.

Because we made RNA-seq from whole termites containing guts, the transcriptome included host termite and symbiont genes. According to the NR species distribution result, there were 22,993 (37.44%) unigenes derived from insect species, which may be supposed to be termite genes, and 38,414 (62.55%) from protozoan symbionts. The distribution result was similar to the study by *Zhang et al. (2012)*. Among termite genes, the majority of the sequences (50.31%) had strong homology with *Zootermopsis nevadensis*, followed by *C. formosanus* (8.22%), *Tribolium castaneum* (3.61%), *Harpegnathos saltator* (3.22%), *Acyrthosiphon pisum* (2.13%) and the remaining species were less than 2% (Fig. 1A). Among symbiont genes, the majority of the sequences (56.72%) had strong homology with genus *Trichomonas*, followed by genus *Toxoplasma* (3.86%) and *Leishmania* (3.37%) (Fig. 1B).

## Transcriptome profiles of worker termites at different $CO_2$ concentrations

Gene expression of all unigenes in T1, T2, T3, and T4 were estimated as FPKM. Genes with FPKMs $\leq$ 1 were considered not to be expressed or to be present at very low levels; genes with FPKMs over 60 were considered to be expressed at a very high level
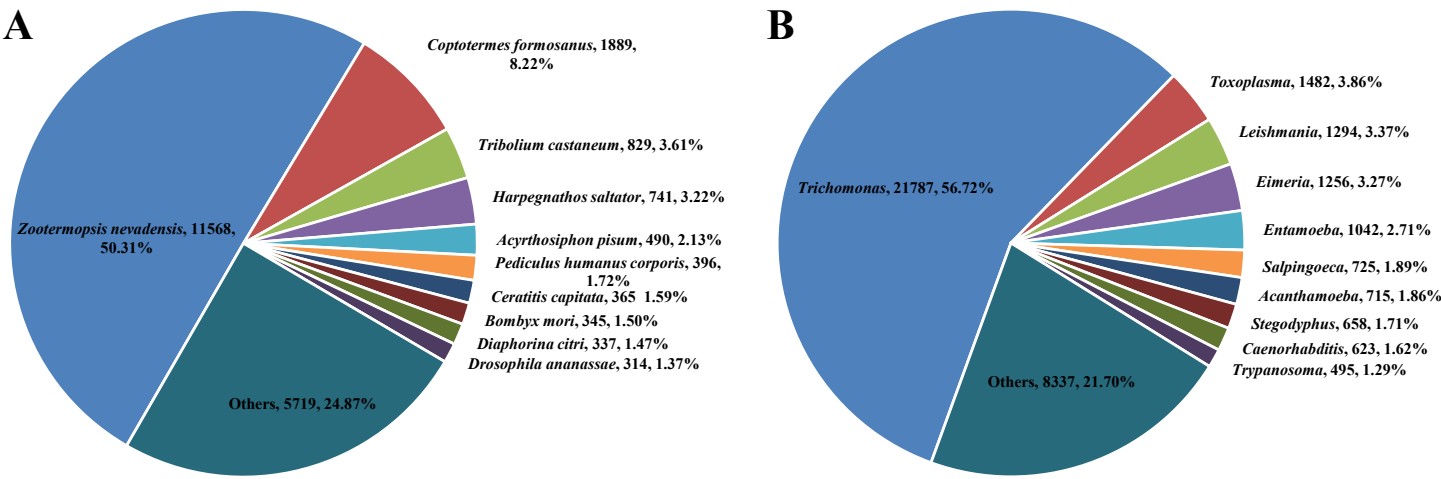

**Figure 1** **Species distribution from BLASTx matches against the NR protein database (cut-off value E < 10⁻⁵).** (A) Species distribution of genes derived from termites and the proportions for each species. (B) Species distribution of genes derived from symbionts and the proportions for each species.

**Table 2** Distribution of gene expression in each $CO_2$ treatments (FPKM >1).

| FPKM interval | T1 | T2 | T3 | T4 |
|---|---|---|---|---|
| 1–3 | 32,255 | 28,433 | 27,736 | 25,713 |
| 3–15 | 14,082 | 13,693 | 13,227 | 12,173 |
| 15–60 | 3,774 | 3,695 | 3,684 | 3,579 |
| >60 | 1,079 | 1,090 | 1,104 | 1,066 |

(*Fang et al., 2015*). Table 2 shows the distribution of expression levels of all genes in each $CO_2$ treatments; the overall trend has been a decline as elevated $CO_2$ concentrations (Table 2). The number of the genes with FPKM >1 shared by T1, T2, T3, and T4 were 24,385 (Fig. S4A) and the four samples had 876 common genes with high expression (FPKM > 60) (Fig. S4B). We analyzed the biological function of the highly expressed genes using the GO (Fig. S2) and KOG classifications. In the GO classification, the most abundant GO terms were 'metabolic process' and 'catalytic activity.' In the KOG classification, these genes were mainly classified into 'translation, ribosomal structure and biogenesis,' 'posttranslational modification, protein turnover, chaperones,' 'cytoskeleton,' and 'energy production and conversion.' Their functions covered metabolism, cellular processes and signaling, and information storage and processing. Thus, these genes may play an essential role in the life of termites. We found that three genes, c155263_c0, c190637_c0, and c188048_c3, were extremely highly expressed (FPKM > 9,000) in all four treatments. Gene c155263_c0 was annotated as a hypothetical protein with unknown function. Gene c190637_c0 was similar to ABC-type transporter Mla, which maintains outer membrane lipid asymmetry and participates in cell wall/membrane/envelope biogenesis. Gene c188048_c3 encoded endo-$\beta$-1,4-glucanase of *C. formosanus*, which is important to termite cellulose digestion system.

**Table 3** The fold change distribution of termite DEGs.

| Pairwise comparison | Variation | Fold change | | | | | Total DEGs |
|---|---|---|---|---|---|---|---|
| | | 2–4 | 4–8 | 8–16 | 16–32 | >32 | |
| T1 vs. T2 | up | 7 | 7 | 6 | 2 | 1 | 23 |
| | down | 30 | 22 | 14 | 12 | 7 | 85 |
| T1 vs. T3 | up | 35 | 65 | 33 | 8 | 4 | 145 |
| | down | 17 | 17 | 13 | 6 | 3 | 56 |
| T1 vs. T4 | up | 71 | 32 | 13 | 11 | 12 | 139 |
| | down | 148 | 148 | 94 | 54 | 10 | 454 |
| T2 vs. T3 | up | 27 | 27 | 31 | 28 | 67 | 180 |
| | down | 18 | 7 | 7 | 4 | 1 | 37 |
| T2 vs. T4 | up | 54 | 50 | 21 | 23 | 23 | 171 |
| | down | 102 | 130 | 92 | 54 | 17 | 395 |
| T3 vs. T4 | up | 74 | 39 | 12 | 7 | 6 | 138 |
| | down | 138 | 137 | 91 | 58 | 32 | 456 |

## Differentially expressed genes (DEGs) and functional annotation

Hierarchical clustering of all DEGs was performed to observe the gene expression patterns based on the $\log_2$ FPKMs for the four samples (Fig. 2). The number of DEGs in each pairwise comparison is presented in Fig. 3. In total, all six comparison sets had 2,936 unique DEGs, 909 were termite DEGs and 2,027 were symbiont DEGs. The number of symbiont DEGs was more than twice greater than the number of termite DEGs, suggesting symbionts changed more remarkably than termite. Approximately 90% DEGs were in comparison sets of T1 vs. T4, T2 vs. T4, and T3 vs. T4, and a majority of them were down-regulated, especially in symbionts. However, in T1 vs. T3 and T2 vs. T3, the number of up-regulated termite DEGs was about twice and four times as many as the number of down-regulated termite DEGs, respectively. Meanwhile, the fold-change of up-regulated termite DEGs was larger than down-regulated termite DEGs in above two comparison sets (Table 3), which suggests genes are slightly up-regulated in T3 in termite but not symbionts.

According to GO classification (Fig. 4), the number of DEGs in some GO terms (e.g., 'oxidation reduction,' 'alcohol metabolic process,' 'ion binding' and 'oxidoreductase activity') was similar between termites and symbionts. But in most GO terms, the number of symbiont DEGs was more than the number of termite DEGs, such as 'cell cycle process,' 'embryonic development,' 'growth,' 'macromolecule localization,' 'transferase activity,' and 'ligase activity.' In KOG classification, the majority of termite DEGs are in the class 'signal transduction mechanisms,' 'lipid transport and metabolism' and 'amino acid transport and metabolism,' while the majority of symbiont DEGs are in the class 'posttranslational modification, protein turnover, chaperones,' 'signal transduction mechanisms', 'translation, ribosomal structure and biogenesis', and 'cytoskeleton.'

Compared to natural $CO_2$ level (T1 vs. T2, T1 vs. T3, and T1 vs. T4), there were 54 common termite DEGs in response to elevated levels of $CO_2$ (Fig. 5A, Table S3).

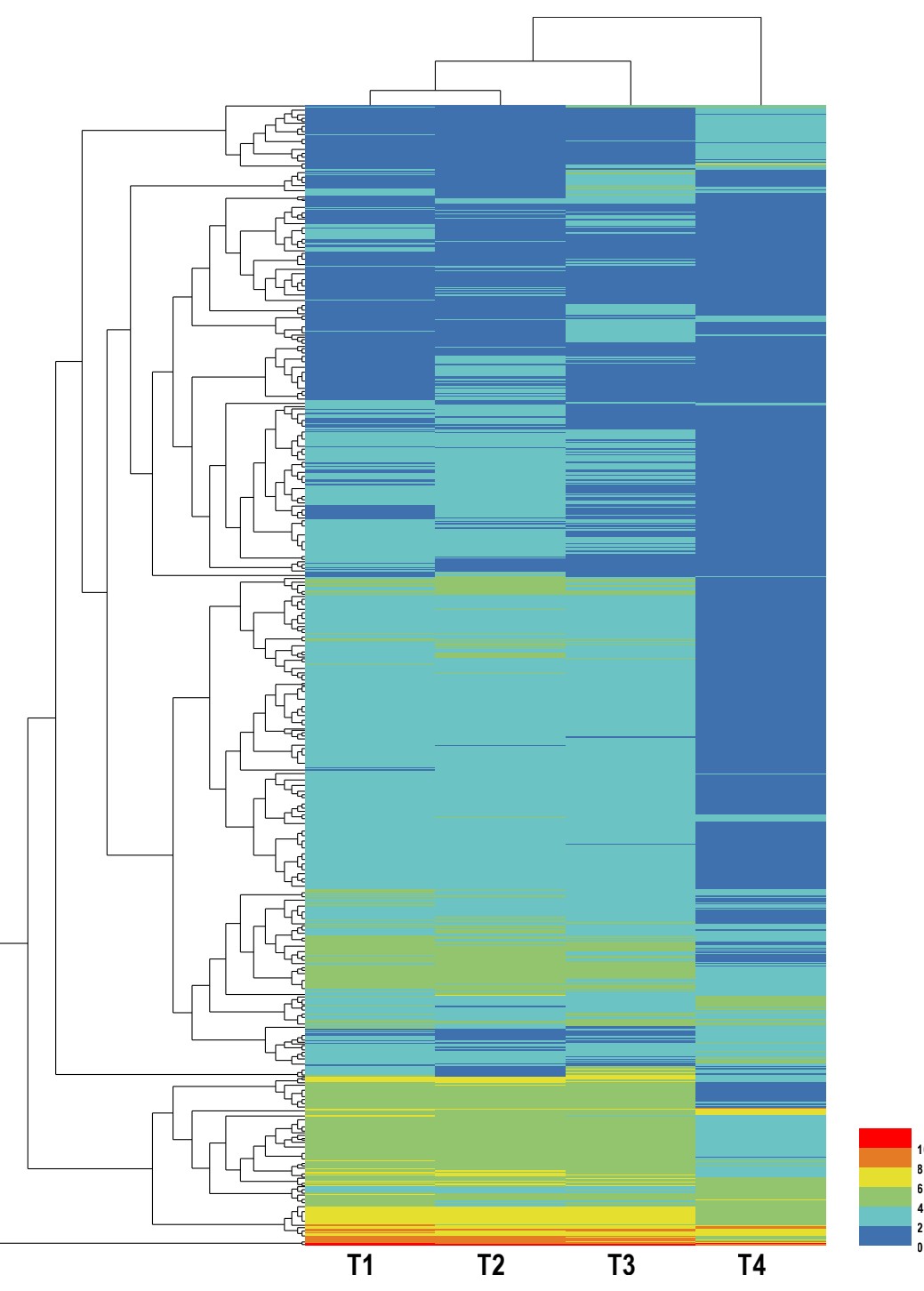

**Figure 2  Hierarchical clustering graph of DEGs between different CO$_2$ treatments.** The blue bands indicate low gene expression quantity; the red bands indicate high gene expression quantity.

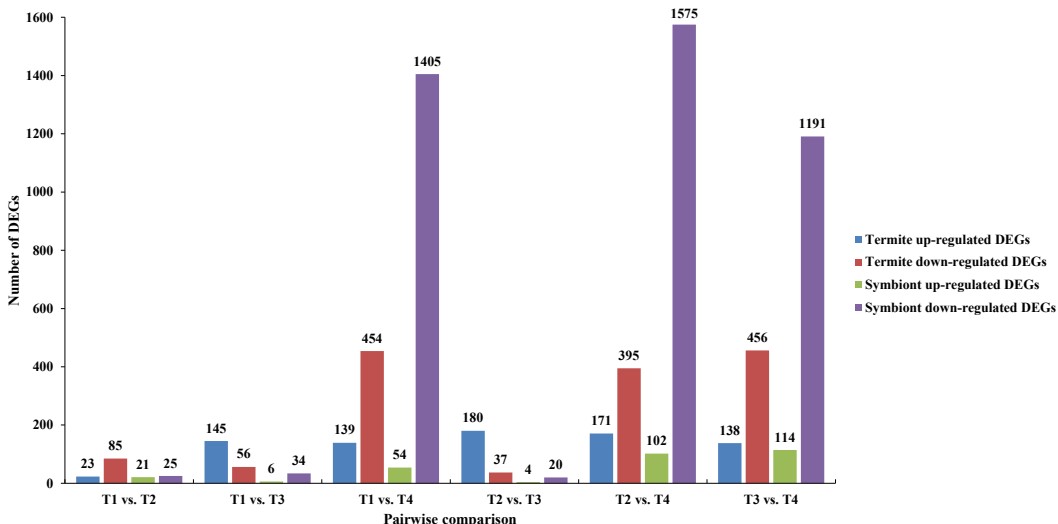

**Figure 3** **Number of differentially expressed genes (DEGs) in each pairwise comparison.** The blue and red bars represented up- and down-regulated DEGs derived from termites, respectively. The green and pink bars represented up- and down-regulated DEGs derived from symbionts, respectively.

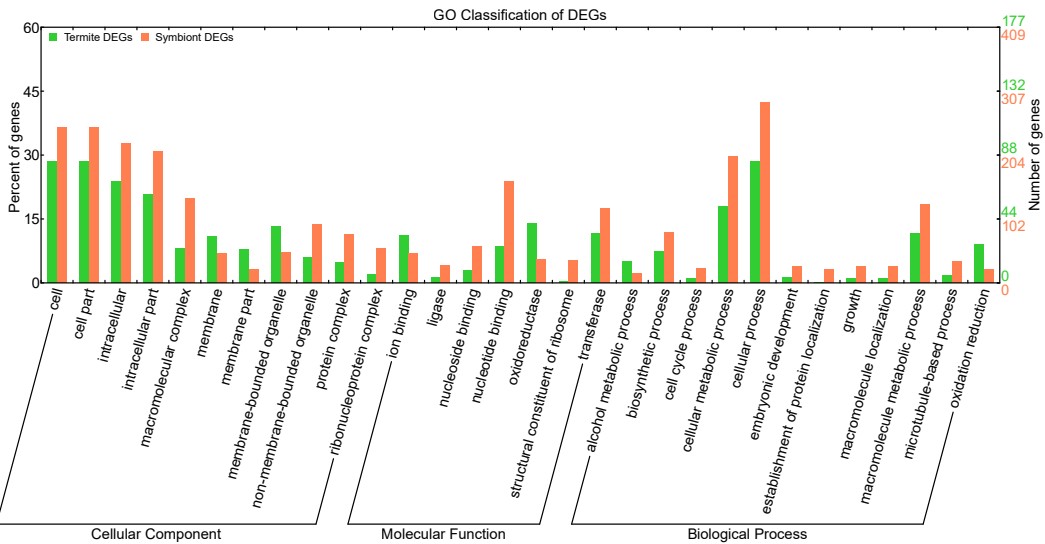

**Figure 4** **Gene Ontology classification of termite and symbiont DEGs.** The green and red bars represented DEGs derived from termites and symbionts, respectively.

Only two DEGs were up-regulated in all three sets. They were annotated as transferrin-like protein (c188927_c0) and prolixicin antimicrobial protein (c127508_c0). Thirty DEGs were down-regulated in all three sets, but only 15 had informative annotations (Table S3). Most of the commonly down-regulated DEGs were annotated as cuticle protein (10 DEGs), which contributes to the structural integrity of a cuticle and takes part in cell wall/membrane/envelope biogenesis. The rest of the commonly down-regulated DEGs included apolipoprotein D, which also participates in cell wall/membrane/envelope

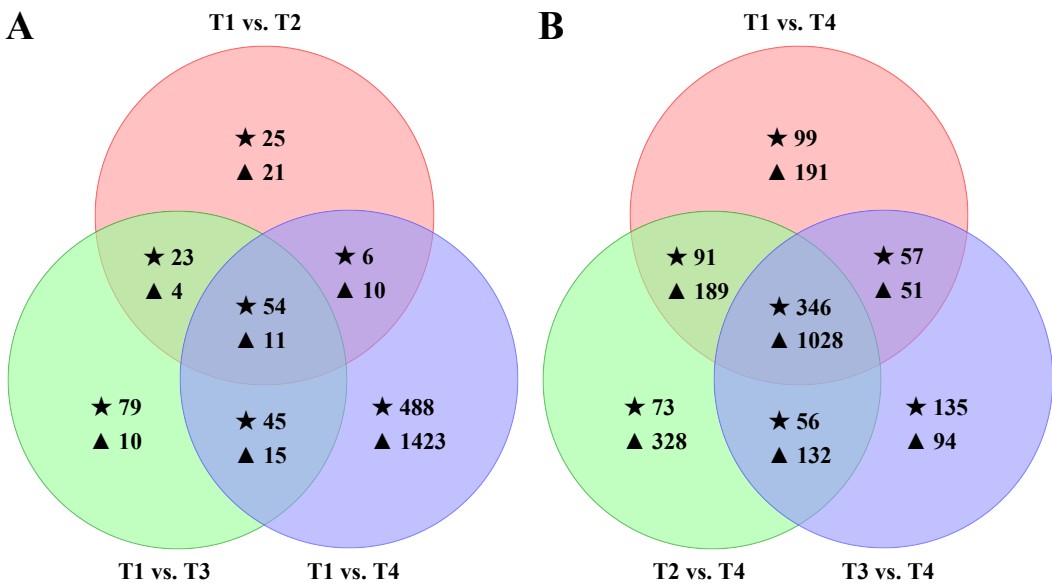

**Figure 5** Effects of the elevated $CO_2$ treatments on the *Coptotermes formosanus* transcriptome. (A) Venn diagram showing the overlaps between the DEGs in elevated $CO_2$ levels and normal air. (B) Venn diagram of the DEGs in T1, T2, and T3 compared to T4. The star (★) represent termite DEGs and the triangle (▲) represent symbiont DEGs.

biogenesis (c102424_c0); collagen precursor, which is involved in extracellular structures (c181121_c0); glucokinase 1, which has transferase activity and participates in cellular metabolic process (c186958_c0); and actin cytoskeleton-regulatory complex protein (c127831_c0 and c169839_c0). Furthermore, 17 common DEGs were down-regulated in T2 and T4 but significantly up-regulated in T3. Among them, ten DEGs were annotated and mainly had three types of function: cuticle protein (c185045_c1, c126213_c0, c174474_c1, c190969_c1), fibroin heavy chain precursor (c128561_c0, c128751_c0, c192228_c0), and period circadian protein (c126015_c0, c174457_c0). For symbiont DEGs, there were 11 DEGs in common (Fig. 5A), 10 of them were down-regulated in T2, T3 or T4 compared to T1, such as c185407_c0 (annotated as cellulase) and c195974_c0 (annotated as ferredoxin-NADP oxidoreductase). Only one gene, c129705_c0 (annotated as threonine dehydratase family protein), was up-regulated in T4.

Compared to high $CO_2$ level (T1 vs. T4, T2 vs. T4, and T3 vs. T4), we found that 346 termite genes were commonly differentially expressed, with 74 up-regulated and 268 down-regulated in all three sets (Fig. 5B). Of the 74 up-regulated DEGs, 41 of them had informative annotations. For example, genes c184494_c2, c105191_c0, and c183958_c0 were highly expressed and associated with lipid transport and metabolism; c173654_c0 was highly expressed and involved in energy production and conversion. Among 268 down-regulated DEGs, 197 received informative annotations, 71 were hypothetical protein or uncharacterized protein. For example, gene c183902_c0, c174002_c0 and c168998_c0 were significantly down-regulated and participated in carbohydrate transport and metabolism. Moreover, we found that most common DEGs were only differential expressed in high $CO_2$ level. A total of 64 up-regulated DEGs did not differentially expressed among T1,

**Table 4** Common enriched GO terms and number of DEGs derived from termites and symbionts.

| GO term | Name | Type[a] | Termite DEGs | Symbiont DEGs |
|---|---|---|---|---|
| GO:0031090 | organelle membrane | CC | 0 | 2 |
| GO:0005743 | mitochondrial inner membrane | CC | 5 | 0 |
| GO:0005737 | cytoplasm | CC | 5 | 38 |
| GO:0016491 | oxidoreductase activity | MF | 15 | 11 |
| GO:0046872 | metal ion binding | MF | 11 | 17 |
| GO:0005506 | iron ion binding | MF | 4 | 2 |
| GO:0020037 | heme binding | MF | 6 | 0 |
| GO:0030246 | carbohydrate binding | MF | 2 | 0 |
| GO:0055114 | oxidation–reduction process | BP | 19 | 21 |
| GO:0006810 | transport | BP | 10 | 20 |

**Notes.**
[a]CC, cellular component; MF, molecular function; BP, biological process.

T2, and T3. For example, two DEGs (c81973_c0 and c174294_c0) were expressed only in T4. Three DEGs (c192331_c0, c180536_c0 and c127808_c0) were not expressed in T1 and T2 but were significantly expressed in T4. However, these five genes were annotated as hypothetical proteins in *Z. nevadensis*. Furthermore, there were 263 down-regulated DEGs that were not differentially expressed among T1, T2, and T3. For example, six genes (c192338_c0, c128228_c0, c129383_c0, c192671_c0, c192511_c0 and c184316_c0) showed high expression in T1, T2, and T3 (FPKM > 15) and low expression in T4 (FPKM < 1). Gene c192338_c0 is annotated as glyceraldehyde-3-phosphate dehydrogenase and plays a role in carbohydrate transport and metabolism. Gene c128228_c0, c129383_c0 and c192671_c0 are ribosomal proteins, which participate in translation, ribosomal structure and biogenesis. For symbiont DEGs, there were 1,028 DEGs in common, with 40 up-regulated and 988 down-regulated in all three sets. Among the down-regulated genes, 64 genes did not expressed in T4 (FPKM = 0), which take part in posttranslational modification, ribosomal structure, or cell wall biogenesis.

## GO and KEGG enrichment analyses of the DEGs

The majority of significantly enriched GO terms were in T1 vs. T4, T2 vs. T4, and T3 vs. T4, specifically more than 130 GO terms enriched in biological process (Table S4). However, only two terms were common in biological process. The common enriched terms and the number of DEGs are listed in Table 4. Both termite and symbiont DEGs were enriched in 'cytoplasm,' 'oxidoreductase activity,' 'metal ion binding,' 'iron ion binding,' 'oxidation–reduction process,' and 'transport.' The termite DEGs were also enriched in 'mitochondrial inner membrane,' 'heme binding,' and 'carbohydrate binding.'

In T1 vs. T2 and T2 vs. T3, the 'oxidative phosphorylation' pathway was significantly enriched and all DEGs were termite DEGs (Table 5). The 'ribosome,' 'glycolysis/gluconeogenesis,' and 'starch and sucrose metabolism' pathways were common enriched in T1 vs. T4, T2 vs. T4, and T3 vs. T4, however, the number of symbiont DEGs was larger than termite DEGs. 'Aminoacyl-tRNA biosynthesis' and 'proteasome' were classified

**Table 5  Significantly enriched pathways in DEGs ($q < 0.05$).**

| Pairwise comparison | KEGG pathway | ko ID | Termite DEGs | Symbiont DEGs |
|---|---|---|---|---|
| T1 vs. T2 | Oxidative phosphorylation | ko00190 | 10 | 0 |
| T1 vs. T4 | Ribosome | ko03010 | 0 | 3 |
| | Glycolysis/Gluconeogenesis | ko00010 | 7 | 15 |
| | Starch and sucrose metabolism | ko00500 | 5 | 11 |
| | Proteasome | ko03050 | 0 | 22 |
| | Aminoacyl-tRNA biosynthesis | ko00970 | 0 | 18 |
| T2 vs. T3 | Oxidative phosphorylation | ko00190 | 10 | 0 |
| T2 vs. T4 | Ribosome | ko03010 | 8 | 62 |
| | Starch and sucrose metabolism | ko00500 | 4 | 12 |
| | Glycolysis/Gluconeogenesis | ko00010 | 7 | 15 |
| T3 vs. T4 | Ribosome | ko03010 | 8 | 65 |
| | Glycolysis/Gluconeogenesis | ko00010 | 7 | 15 |
| | Starch and sucrose metabolism | ko00500 | 4 | 11 |
| | Proteasome | ko03050 | 0 | 22 |
| | Aminoacyl-tRNA biosynthesis | ko00970 | 0 | 19 |

in the KEGG 'genetic information processing' category, and were common enriched in T1 vs. T4 and T3 vs. T4. Both were changes of symbionts.

## Expression profiles of chemosensory proteins

According to annotations and conserved protein domains, two ORs, five GRs, four IRs, 22 OBPs, and two CSPs were identified by the 7tm Odorant receptor (cl20237), 7tm chemosensory receptor (pfam08395), PBP2_iGluR_putative (cd13717), PBP/GOBP family (pfam01395), and insect pheromone-binding family OS-D (pfam03392) domain, respectively. Among these 35 genes, eight genes had a relatively high expression in at least one library (FPKM > 10), and most of them were up-regulated in T3 (Table S5). Six OBPs (c110031_c0, c128738_c1, c129041_c0, c192285_c0, c192783_c0, and c193269_c0) were significantly up-regulated in T3 compared to T1. One OBP, c128814_c0, was significantly increased in T3 compared to T2. One CSP, c125410_c0, was significantly increased in T3 compared to the other three libraries.

## Validation of RNA-seq data by qRT-PCR

To validate the transcriptome result, we selected 10 DEGs for qRT-PCR confirmation (c125410_c0, c129041_c0, c166756_c0, c167200_c0, c168998_c0, c169342_c0, c173654_c0, c179746_c0, c181311_c0, and c184494_c2, five genes were described in the text). The primers used for qRT-PCR were shown in Table S1. The amplification efficiency of each primer set was validated; standard curves (10× serial dilutions) yielded regression lines with $R^2$ values > 0.97 and an amplification efficiency ranging from 0.9–1.1 (ideal value of 0.8–1.2). Each primer set produced a single amplicon as judged by the single peak in the dissociation curve. The qRT-PCR expression results (Fig. S5) were similar to the results obtained from the Illumina sequencing data. Three DEGs were highly expressed in T2

in the transcriptome results but minimally expressed in the qRT-PCR results. Although the expression levels were not completely consistent (possibly due to different methods of library construction, reference genes, normalization, or biological differences), the results fundamentally supported the reliability of the RNA-seq results.

## DISCUSSION

### Overview of transcriptome data

*C. formosanus*, a worldwide important pest, has been studied extensively in omics, including genome, transcriptome, metabolome, DNA methylome, and 16S rRNA sequencing (*Scharf, 2015*). While most studies have focused on symbionts, a few have combined host and symbiont, considering the whole termite (*Scharf, 2015*). Those studies are mainly based on conventional Sanger sequencing; rarely has Illumina high-throughput sequencing study been reported to date. Compare to the study by *Zhang et al. (2012)* using Sanger sequencing, the present study newly assembled transcriptome contains massive amounts of data (11.02 GB) using Illumina sequencing, and covers different developmental stages and castes (larva, worker, pre-soldier, soldier, reproductive). The genetic information will facilitate future developmental and caste differential studies of *C. formosanus*, and contribute to future work in termite comparative genomics.

### Transcriptomic response to elevated $CO_2$ treatments

In this study, we exposed workers of *C. formosanus* to 0.04%, 0.4%, 4%, and 40% $CO_2$ concentrations and constructed four transcriptomes to examine the gene expression profiles. Hierarchical clustering of all DEGs showed that the expression patterns of T1, T2, and T3 were very close, particularly T1 and T2; some DEGs were increased in T3; and more than one-third of DEGs showed reduced expression in T4 (Fig. 2). Since termites were collected and placed in a sealed container for 72 hr, the final $CO_2$ level was higher than the initial concentration, which was 0.85% ± 0.07%, 1.11% ± 0.01%, 4.67% ± 0.01%, and 40.61% ± 0.04%, respectively. The order of the final $CO_2$ concentration levels was still T1 < T2 < T3 < T4. However, the final T1 concentration was close to T2, which may result in the similar expression pattern of T1 and T2 (Fig. 2). The majority of the *C. formosanus* lifetime is spent living inside wood. The $CO_2$ concentration in the nest, which was similar to the T3 treatment, is higher than it outside the nest. When termites go outside the nest, it is similar to the T1 or T2 treatment. Termites have adapted to a life in the nest or in enclosed galleries and are prone to perish quickly when exposed to the open atmosphere (*Stange & Stowe, 1999*). To some extent, this may be influenced by $CO_2$ concentration, which may carry information relevant to termites, such as information on the location of their nest (*Stange & Stowe, 1999*). Thus, termites may increase gene expression and fit better in T3 treatment. The 40% of $CO_2$ was abnormally high and some termites were dead after 72 h. Although we collected live termites for experiment, we cannot rule out the possibility that termites were damaged by $CO_2$ exposure, suggesting that some changes in gene expression may be not directly associated with the $CO_2$ effects. We also noted that symbionts, intestinal protists and bacteria, accounted for the majority of changes (69% DEGs derived from symbionts) and their expression mainly decreased in T4. Because high concentrations

of $CO_2$ might affect pH in the termite guts, and cause changes in intestinal flora. It is likely that the protists were killed by the abnormally high $CO_2$ level, and as a result, gene expression levels of them were depressed. The death of protozoans may be $CO_2$ direct effect, or combined effects of $CO_2$ and other general stresses. However, the comparisons of transcript levels employed in our study are based on the assumption that total RNA content per cell remains constant. *Lin et al. (2012)* recently found transcriptional amplification in tumor cells with elevated c-Myc level, and *Lovén et al. (2012)* further indicated that many up-regulated DEGs were missed and down-regulated ones were falsely produced when processed by global normalizations. The extent to which this will force reconsideration of present expression studies is as yet unclear, especially the down-regulated DEGs. This problem will still be studied in the future.

To help understand the $CO_2$ effects on termite biological processes and gene functions, termite DEGs were analyzed using the public databases. The over-represented GO terms were evaluated to infer which molecular functions, cellular components and biological processes were most affected by the experimental conditions (Table 4). For molecular function, elevated $CO_2$ levels influenced oxidoreductase activity, metal ion binding, iron ion binding, heme binding, and carbohydrate binding. From studies in *Drosophila* and other insects, the receptors used to recognize olfactory stimuli appear to be ion channels, which may be associated with the enrichment of ion binding terms (*Spehr & Munger, 2009*). For the biological process, oxidation–reduction process and transport were affected, which may be linked to anaerobic respiration (*Nielsen & Christian, 2007*). Studies showed that gene expression may be suppressed to reduce oxygen, aerobic and metabolic activities, including oxidative phosphorylation, oxidation–reduction process, and carbohydrate metabolism in extremely high $CO_2$ concentrations (*Nielsen & Christian, 2007*). From the KEGG enrichment results, we found that high $CO_2$ levels significantly influenced ribosome, glycolysis/gluconeogenesis, and starch and sucrose metabolism pathways (Table 5). Briefly, there were three aspects effected by elevated $CO_2$: (1) carbohydrate metabolism, such as the binding process, and substrates such as glucose, starch and sucrose; (2) energy metabolism, such as genes with oxidoreductase activity that take part in oxidation–reduction process and the oxidative phosphorylation pathway; and (3) the directed movement of substances (such as metal ion, iron ion, heme, and carbohydrate) by means of some agent such as a transporter or pore.

## Genes associated with chemosensory system

In insect chemosensory systems, three chemosensory receptor multi-gene families (ORs, GRs, and IRs) are involved in detection, while OBPs and CSPs play a role in recognition (*Brand et al., 2015*). OR and GR proteins are highly diverse, with many sharing only 20% and 8% amino acid similarity, respectively (*Hallem, Dahanukar & Carlson, 2006*). The extraordinary divergence in sequences makes it difficult to detect and discriminate *OR* and *GR* genes by traditional sequencing methods. Insect *OR* and *GR* genes were first discovered in the genome sequence of *Drosophila melanogaster*, suggesting that these genes could largely be discovered from genome sequences. Thus, the transcriptome of *C. formosanus* may provide information on the candidate chemosensory genes. Totally,

two *ORs*, five *GRs*, and four *IRs* were identified. The number of *OR* genes was obviously smaller than that of other insects, such as *D. melanogaster*, *Anopheles gambiae*, and *Apis mellifera* which have 60, 79, and 170 *OR* genes, respectively (*Robertson & Wanner, 2006*). One *OR*, c197137_c0, was homologous to Or83b of *Holotrichia oblita*, *Plutella xylostella*, *Helicoverpa assulta*, etc. Or83b is highly conserved among all insect species analyzed so far (*Nakagawa et al., 2012*). The number of *GR* genes was close to *Ap. mellifera* which has 10 *GR* genes, while *D. melanogaster* and *An. gambiae* have 60 and 79 *GR* genes, respectively (*Robertson & Wanner, 2006*). However, five GRs were not homologous to *D. melanogaster* GR21a or GR63a, and their expression was not significant under $CO_2$ stress. Perhaps, *GRs* in *C. formosanus* do not act as $CO_2$ receptors. However, we note that it is unlikely that the detected candidate genes represent the complete repertoire of the *C. formosanus* chemosensory gene families because detection is not possible if expression levels of target genes are too low or if they are specific to unexamined sexes, castes, life stages or tissues (*Brand et al., 2015*). The detected genes are likely important and typically among the highest expressed chemosensory genes in *C. formosanus* and thus are very likely to be detected in transcriptome analyses. However, more chemosensory genes and their functions should be examined in further experiments.

The two non-receptor multi-gene families, OBPs and CSPs, encode soluble proteins and have been identified in the lymph of chemosensilla and non-sensory organs in insects (*Pelosi, Calvello & Ban, 2005*). They contribute to the transport of odorant molecules through sensillar lymph, and increase the sensitivity and possibly the selectivity of the insect olfactory system (*Leal, 2013*). OBPs are reported to be different across species and within the same species, sharing even less than 20% amino acid identity in some cases (*Zhou, Field & He, 2010*). The number of *OBP* genes in different insects ranges from 15 (*Acyrthosiphon pisum*) to 66 (*An. gambiae* and *Aedes aegypti*) (*Fan et al., 2011*). In this study, we identified 22 *OBP* genes. According to their putative protein sequences, these OBPs could be divided into two groups: 11 were classical OBPs with six cysteine residues (Fig. 6A), and 11 were Minus-C OBPs with four or five cysteine residues (*Fan et al., 2011*; *Pelosi et al., 2006*). Among them, seven *OBP* genes were differentially up-regulated in T3, including five classical OBPs and two Minus-C OBPs. OBPs may perform roles either related or not related to chemoreception, as they are widely distributed throughout the insect's body, including different sensory parts (e.g., antennae and mouth), tarsi and wings (*Pelosi et al., 2006*). However, the expression of receptor genes was inconsistent with *OBP* genes, which makes it difficult to explain. Both the response of *OBP* genes to elevated $CO_2$ levels and the downstream response elements require more experiments. CSPs are smaller than OBPs and present a motif of four conserved cysteines (*Angeli et al., 1999*). The number of CSPs reported in each species is quite variable, such as *Cactoblastis cactorum*, *Polistes dominulus*, and *Vespa crabro*, with only one CSP, *D. melanogaster* with four CSPs, *An. gambiae* with seven CSPs, and *Locusta migratoria* with at least 20 CSPs (*Pelosi et al., 2006*). Here, we identified two *CSP* genes based on their sequences (Fig. 6B), and c125410_c0 was differentially up-regulated in T3. Some CSPs, such as CLP-1 of *Cactoblastis cactorum* and OS-D of *Drosophila*, have been reported to be involved in the perception of carbon dioxide

**A**

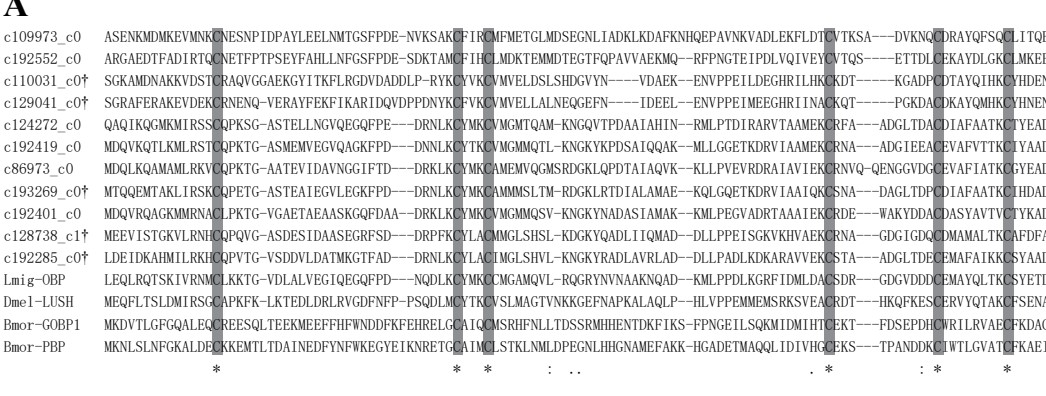

**B**

Figure 6 **Alignment of the partial amino acid sequence of *Coptotermes formosanus* OBPs and CSPs with that from other insects.** Grey boxes show conserved cysteines. (A) Alignment of eleven *C. formosanus* putative classical OBPs with other insects. The symbol † represent DEGs. (B) Alignment of two *C. formosanus* putative CSPs with other insects.

(*Maleszka & Stange, 1997*; *McKenna et al., 1994*). Thus, the up-regulation of c125410_c0 may be in response to increased carbon dioxide.

Odorant stimulation of olfactory receptor neurons results in a calcium influx modulating signal transduction pathways (*Ronnett & Moon, 2002*). Therefore, elevated $CO_2$ levels may affect elements in signal transduction pathways. According to KEGG annotation, we found that four genes annotated as calmodulin in the olfactory transduction pathway (ko04740) were significantly down-regulated in T4 compared to T1, T2 and T3. Among them, calmodulin c181311_c0 had the highest FPKM value in all libraries, indicating that it is a major, common gene in the pathway. It encodes for a protein of 170 amino acids, characterized with two EF-hand or calcium binding motifs. There is evidence in the literature that the inhibition of calmodulin gene expression eliminates the $CO_2$ gating sensitivity of connexin channels (*Peracchia et al., 2003*). Our results show that high $CO_2$ concentration significantly suppresses calmodulin gene expression, while medium and low $CO_2$ concentration have slight effect on the gene. However, the link between $CO_2$ and calmodulin as well as the underlying mechanism need more experiments to illustrate.

## CONCLUSION

Overall, we have identified 2,936 genes with dynamic regulation under elevated $CO_2$ conditions belonging to diverse pathways, mainly metabolic processes and signal transduction. The candidate chemosensory proteins were also identified in *C. formosanus*, and some of them likely play a role in $CO_2$ sensing. This preliminary study provide a

number of candidate genes that may be used as starting point to dissect the gene regulatory network involved in termite responses to $CO_2$.

## ACKNOWLEDGEMENTS

The authors thank the Biomarker Biotechnology Corporation (Beijing, China) for assisting with transcriptome sequencing. We are grateful to all people who in any way contributed to the development of this work.

### Funding

This research was supported by the National Natural Science Foundation of China (NSFC) (31172163), Funds for Environment Construction & Capacity Building of GDAS' Research Platform (2016GDASPT-0107), and Science and Technology Planning Project of Guangzhou city, China (201510010036). The funders had no role in study design, data collection and analysis, decision to publish, or preparation of the manuscript.

### Grant Disclosures

The following grant information was disclosed by the authors:
National Natural Science Foundation of China (NSFC): 31172163.
Environment Construction & Capacity Building of GDAS' Research Platform: 2016GDASPT-0107.
Science and Technology Planning Project: 201510010036.

### Competing Interests

The authors declare there are no competing interests.

### Author Contributions

- Wenjing Wu conceived and designed the experiments, performed the experiments, analyzed the data, contributed reagents/materials/analysis tools, wrote the paper, prepared figures and/or tables, reviewed drafts of the paper.
- Zhiqiang Li conceived and designed the experiments, analyzed the data, wrote the paper, reviewed drafts of the paper.
- Shijun Zhang contributed reagents/materials/analysis tools.
- Yunling Ke reviewed drafts of the paper.
- Yahui Hou performed the experiments.

### DNA Deposition

The following information was supplied regarding the deposition of DNA sequences:

All sequence data have been submitted to GenBank Sequence Read Archive databases under accession number SRP068272 and SRP068332, and associated with Bioproject PRJNA308390 and PRJNA308507, respectively. Their accessions are SRR3095926 for Cfo (reference transcriptome of *C. formosanus*), SRR3097983 for T1, SRR3097984 for T2, SRR3097985 for T3, and SRR3097987 for T4.

## Data Availability

Transcriptome assembly of *Coptotermes formosanus*:

Wu W, Zhiqiang L. 2016. All combination unigenes. Figshare: https://dx.doi.org/10.6084/m9.figshare.3444866.v1;

Transcriptome annotations of *Coptotermes formosanus*: Wu W, Zhiqiang L. 2016. All database annotation. Figshare: https://dx.doi.org/10.6084/m9.figshare.3443930.v1.

## Supplemental Information

Supplemental information for this article can be found online at http://dx.doi.org/10.7717/peerj.2527#supplemental-information.

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
