# Peer review of "Transcriptome response to elevated atmospheric CO2 concentration in the Formosan subterranean termite, Coptotermes formosanus Shiraki (Isoptera: Rhinotermitidae)"

_PeerJ, doi:10.7717/peerj.2527_

## Round 0.1 · original submission · Minor Revisions

Dear Dr. Wenjing Wu,

I am writing to inform you that your manuscript needs some revisions prior to publication in PeerJ.

My personal opinion is that this is a quite interesting manuscript and it will likely be suitable for publication in PeerJ if it is revised to address the points below. The major shortcoming found in the paper concerns the materials & methods and in particular the level of CO2 of the control group that should be higher than 0,04% because the specimens were placed in a sealed container. This has to be taken into account in the discussion.

Please, also follow carefully the suggestions of the Reviewer 1 about KEGG database use.

Reviewer 1 ·

Basic reporting

No major problems. A couple of little typos:

- Line 30 (Abstract. Results): "presented a discontinuous changes" >> "presented discontinuous changes".

- Line 273: "remarkable" >> "remarkably"

Experimental design

Methods and tools used in this paper are standard and established ones. No problems here.

Minor point:

- Line 178: KEGG pathway database is species-specific, and apparently there is no entry for the species studied here on KEGG. Please explain exactly how you mapped DEGs found here to KEGG items.

Validity of the findings

The data seems generally consistent.

Minor points:

- Figure S5: Any idea why T2 shows more discrepancy between RNA-seq and qRT-PCR results than other samples?

- Since the authors believe high CO2 concentration (40%) slows down metabolism generally, discuss how the lack of external standards may have affected the inferences made here (e.g. look at Loven et al. 2012 "Revisiting global gene expression analysis").

Reviewer 2 ·

Basic reporting

Good information provided with nice introduction though it should be noted that the CO2 attraction shown by Bjostad's lab in 2005 has been commercialized and is used in Ensystex bait systems under the name Focus. Discussion could be shortened by not repeating introductory information or results.

Experimental design

Problems exists in the materials and methods. We are told that all castes and developmental stages were examined with regards to CO2 expression changes but only results from a combination of workers and soldiers is reported. More importantly there is no information on the CO2 level from the "control" groups that served as a basis for examining changes based on specific levels of CO2 to termites for 72hr. Since termites were collected and placed in a sealed container for some unspecified time before testing the CO2 level would have certainly been higher than ambient CO2. Possibly the researchers can replicate the time lines and test with the CO2 meter to know what the baseline group was. It appears that only one group of termites was collected so any replication for gene expression was from that one colony group only.

Validity of the findings

As noted above we cannot substantiate the findings of low or medium CO2 levels from baseline data since we don't know what baseline data groups were under with regards to CO2 levels. Thus, we are at the mercy of the authors when it is stated that suppression of gene expression is found from low and high CO2 concentrations (line 408-409). It appeared clear that at 40% CO2 levels protozoans were killed but the authores state that perhaps they died for an unspecified reason. In which case, the results as presented are suspect to environmental changes that were not tested. They again state that they obtained massive amounts of data on larva, worker, pre-soldiers and soldiers and reproductives but all that they report on is a combined group of workers and some soldiers. The conclusion does appear more in line with their research in that this study provides genetic [information] on gene expression in Formosan termites at different CO2 levels and that the results are as they state preliminary.

---

## Round 0.2 · Minor Revisions

Dear Dr. Zhiqiang Li

I appreciate the effort you put in answering to the reviewers’ comments.

However my feeling is that the recommendation made by the reviewer regarding the level of CO2 of the control group was only partially addressed.

In my opinion, the sentence in the discussion on the higher increment of CO2 in T1 than T2, T3 and T4 is not clear: “The increasing amount of CO2 level was similar among different treatments. However, relative to initial concentrations, the increment mostly affect T1 rather than T2, T3 and T4” (lines 401-402).

According to this sentence, it seems that the authors know the final CO2 concentration of T1, T2, T3 and T4. If this is true, it must be added in the manuscript.

Indeed, the measurement of the final CO2 concentration would substantially improve the manuscript.

---

## Round 0.3 · accepted · Accept

I think that the revised manuscript is now considerably improved
compare to its initial version and is suitable for publication in PeerJ.